# Adjustable Vibration Exciter Based on Unbalanced Motors

**DOI:** 10.3390/s23042170

**Published:** 2023-02-15

**Authors:** Volodymyr Osadchyy, Olena Nazarova, Taras Hutsol, Szymon Glowacki, Krzysztof Mudryk, Andrzej Bryś, Anatolii Rud, Weronika Tulej, Mariusz Sojak

**Affiliations:** 1Department of Electric Drive and Automation of Industrial Equipment, Zaporizhzhia Polytechnic National University, Zhukovsky, 64, 69-063 Zaporizhzhia, Ukraine; 2Department of Mechanics and Agroecosystems Engineering, Polissia National University, 10-008 Zhytomyr, Ukraine; 3Department of Machine Use in Agriculture, Dmytro Motornyi Tavria State Agrotechnological University, 18, 72-000 Zaporizhzhia, Ukraine; 4Department of Fundamentals of Engineering and Power Engineering, Institute of Mechanical Engineering, Warsaw University of Life Sciences (SGGW), 02-787 Warsaw, Poland; 5Faculty of Production and Power Engineering, University of Agriculture in Krakow, 30-149 Krakow, Poland; 6Faculty of Engineering and Technology, Higher Educational Institution “Podillia State University”, 32-300 Kamianets-Podilskyi, Ukraine

**Keywords:** vibration exciter, unbalance, angular position, mathematical model, control system, electromechanical system, electric drive

## Abstract

In European industry, such as metallurgical, mining and processing, construction, food, and chemical, vibration exciters are used, which indicates their wide and, in some cases, unique technological capabilities. The most common are electromagnetic and unbalanced vibration exciters. The advantages of electromagnetic vibration exciters include the ability to control the amplitude of the vibration by changing the electrical power supplied; the disadvantages are high material consumption. However, unbalanced vibration exciters have low energy efficiency, which is associated with difficult start-up conditions and with an overestimated mechanical power of the vibration exciter in relation to the power required by the technology itself, which is due to the need to minimize the effect of the technological load on the operating mode of the vibrating unit. Adjusting the amplitude of the disturbing force of unbalanced vibration exciters, regardless of the vibration frequency, will make it possible to reduce the installed power of the unit by passing the resonant frequency with a minimum disturbing force and compensating for the effect of the process load by means of a closed-loop electric drive. In the course of the study, an analytical description of the interaction of the rotating unbalances located on a common movable platform was obtained. On the basis of these analytical dependencies, a mathematical model was developed that takes into account the dynamic characteristics of a frequency-controlled asynchronous electric drive of a closed-loop control system for the mutual arrangement of rotating unbalances. The simulation results confirmed the possibility of using the specified electric drive to control the oscillation amplitude directly in the process of operation of a four-unbalanced vibration exciter. A physical experiment was carried out to determine the transient processes of changing the angular velocity of an induction motor with an abrupt change in the frequency converter setting. On the basis of this experiment, the previously created mathematical model was refined in terms of describing the dynamic parameters of the electric drive. The proposed structure of the control system, the performance of which has been confirmed by mathematical modeling, makes it possible to implement an adjustable four-unbalanced vibration exciter using single commercially available asynchronous vibrators.

## 1. Introduction

Unbalanced and electromagnetic vibration exciters [1] are used quite often in various industrial sectors of Europe [2,3] due to the fact of their wide technological capabilities. However, both types have disadvantages; in particular, electromagnetic ones have a high material consumption, and mass produced unbalanced ones do not allow for changing the amplitude of the disturbing force during operation and have low energy efficiency [4]. The use of the previously proposed adjustable four-weighted vibration exciter makes it possible to reduce the material consumption while providing the possibility of adjusting the amplitude of the disturbing force.

The regulation of the vibration parameters in conjunction with the automatic control of the operating modes of the vibrating units allow for intensifying the technological processes and improving the quality of products.

Increasing the energy efficiency of the vibrating units, as shown in [5], as an example of unbalanced vibration exciters, is possible by using controlled vibration machines. Based on the analysis [6] and classification of systems of controlled vibration drives, it can be seen that kinematically complex mechanisms are used for unbalanced vibration exciters, which entails an increase in the cost of the product and an increase in the cost of its operation.

Therefore, the creation of a control system for a four-unbalanced vibration exciter, which provides regulation of the amplitude of the disturbing force, in order to reduce the power of the used vibrating units and increase their energy efficiency, is an urgent task.

The aim of the work was to improve the energy efficiency of an unbalanced vibration exciter by developing a control system for a four-unbalanced vibration exciter that provides regulation of the amplitude of the disturbing force by changing the mutual angular position of the rotating unbalances. 

A large number of studies have been devoted to the issue of the energy-efficient control of electric drives and electromechanical systems, the results of which have been implemented in various fields of science, technology, and economy [7,8,9,10]. As is known [11], when creating energy-efficient vibration machines, the influence of the technological load and its correct mathematical description come to the fore; however, in modern works [12,13] devoted to the dynamics of the vibration equipment, with a sufficiently detailed mathematical description of the mechanical part of the technological load is taken into account in a very simplified way. Therefore, in terms of energy saving, complex studies are relevant, taking into account not only the direct effect of the drive on the technological load, which is true for an unlimited power source but also the reverse effect of the technological load on the controlled electric drive. In fact, the task of a closed-loop controlled electric drive of vibration machines with unbalances is to ensure the synchronous rotation of the mechanisms with their given relative position.

The study of the process of synchronization of unbalanced vibration exciters due to the fact of an interaction through a movable platform is described in detail in [14], but, according to the authors of the article, not enough attention is paid to the influence of the technological load on the synchronous operation mode. At present, there are practically no studies in the field of ensuring the synchronous operation of the mechanisms of vibration machines by means of a closed-loop controlled electric drive, the synthesis of regulators, and the search for the boundaries of the stability of the synchronous mode, without which it is impossible to create highly efficient vibration machines. References [15,16] are devoted to the study of the possibility of using a closed-loop adjustable electric drive to ensure the synchronous rotation of unbalances; however, the problems of passing the resonance zone and controlling the relative position of the unbalances and compensating torques when the axes of rotation of the unbalances do not receive proper further development.

It was shown in [17] that the use of four unbalances makes it possible to compensate for the rotational moments caused by the mismatch of the axes of rotation of the unbalances.

Given the limitations of the traditional feeder drive mode when transporting bulk materials, ref. [18] proposes a new scheme for an ultrasonic vibrating feeder with a double symmetrical axial structure based on the existing traveling wave ultrasonic transport device.

The dynamics of a vibrating machine with an unbalanced vibration exciter was studied, taking into account the elasticity of its connection with an asynchronous motor. With the help of computer modeling, the occurrence of resonant torsional oscillations of the drive of the vibrating machine during the passage of the region of natural frequencies and the associated manifestation of the Sommerfeld effect were demonstrated. The effectiveness of the proposed recommendations for reducing drive oscillations has been con-firmed [19].

The work in [20] investigated the dynamic characteristics of a vibrating machine with two vibrating elements. The vibrating machine was modeled as a discrete-continuous system. The equations of the motion were developed, and the main parameters of the vibrating machine were determined. The criteria of the first and second stability ranges of the vibrating machine as a resonant system were defined.

The use of mathematical and computer modeling is one of the most common tools for organizing research in almost any field [21,22,23]. The combination of various modeling methods and means opens up new opportunities for researchers and expands the potential of studying the peculiarities of electromechanical and electromagnetic processes in the studied objects [24,25,26,27]. Complex interrelated objects of research are often difficult to describe mathematically, so to do this, various principles and methods of identification are used [28,29,30]. Taking into account the implementation of the principles of Industry 4.0, the remote control of any technological process must be organized taking into account the requirements for cybersecurity and data exchange security [31,32,33]. The generalized models of one-, two-, and three-mass vibrating machines with rectilinear progressive movement of the platforms and a vibration exciter in the form of a ball, roller, or pendulum autobalancer were built. In the generalized model of a single-mass vibrating machine, the platform rests on an elastic–viscous resistance with guides that provide rectilinear translational movement of the platform [31]. Reference [32] proposes a study of the instantaneous electric power of the engine for the evaluation of the mechanical vibrations of the electromechanical complex. With the help of mathematical modeling, a study of the effect of mechanical vibrations of the gearbox on the instantaneous electric power of an asynchronous drive motor was carried out. To ensure a more efficient separation process, a new type of vibrating drum separator was proposed. Based on the asymptotic methods of nonlinear mechanics, a nonlinear mathematical model was built that describes the dynamics of the separator. It can be used to design and select the operating modes of separators of various types with a vibration drive, as well as for other vibration machines [33]. Reference [34] presents a new algorithm for the vibration optimization for a microelectromechanical systems accelerometer without sensor fusion. The experimental results show that the proposed algorithm always provided smaller variations than the low-pass filter, approximately 0.2 degrees of standard deviation. Reference [35] proposed and tested a piezoelectric actuator based on three sandwich longitudinal vibrators. Two horizontal vibrators were parallel to each other and orthogonal to the vertical vibrator. The ends of the two horizontal vibrators served as four driving legs. The mechanical characteristics of the prototype showed that the excitation voltages could be used to control the speed due to the fact of an approximately linear relationship between them.

It should be noted that with the external simplicity of the considered adjustable vibration exciters, the calculation of their dynamics and implementation in practice is complicated by the mutual influence of electric drives and the significant nonlinearity of their relationship. In addition, the impossibility of a detailed analysis of a wide range of destabilizing factors does not allow for the use of classical methods for the synthesis of controllers, which requires a special approach that allows you to create a control part of the system that provides the required accuracy and speed with incomplete a priori information about the parameters and properties of the object management.

In unregulated vibration exciters, vibration motors [36] are widely used, which use asynchronous motors as drives. To determine the relative position of the rotating unbalances of the previously mentioned mass produced vibration motors, special sensors are required that combine high accuracy and vibration resistance at a low cost compared to the cost of the entire vibration unit. One of the solutions is the sensor proposed in [37].

It is advisable to use commercially available frequency converters as devices that regulate the rotational speed of unbalances. The latter are characterized by a wide range: from highly dynamic electric drives [11], which are practically not inferior to DC drives, to low-cost converters that provide strict dependencies between the output frequency and voltage.

The use of mass produced vibration motors and frequency converters will reduce the time and cost of development, as well as speed up the transition from a prototype to implementation in production.

Therefore, the development of a four-balance adjustable vibration exciter based on commercially available vibration motors with frequency control is an urgent task.

The aim of this work was to develop an adjustable four-weighted vibration exciter based on an asynchronous frequency-controlled drive with a position feedback sensor and to study the electromechanical processes of the specified vibration exciter.

A description of the method of conducting the research follows. In the course of the study, an analytical description of the interaction of the rotating unbalances located on a common movable platform was obtained. On the basis of these analytical dependencies, a mathematical model was developed that takes into account the dynamic characteristics of a frequency-controlled asynchronous electric drive of a closed-loop control system for the relative position of rotating unbalances. The simulation results confirmed the possibility of using the specified electric drive to control the amplitude of the oscillations directly in the process of the operation of a four-unbalanced vibration exciter.

In the Section 2 (Materials and Methods), the physical and mathematical content of the idea of managing four imbalances is described. Section 3 (Results) is devoted to the development of a computer model for the study of a control system of a four-unbalanced vibration drive and the analysis of the results of the physical and computer modeling. Section 4 (Discussion) provides a detailed description and comparison of the proposed methodology with others presented in the literature. Section 5 contains the conclusions of the article.

## 2. Materials and Methods

The amplitude of the disturbing force of the unbalanced vibrators can be set from zero to the maximum value by changing the relative position of the unbalances from 180° to 0°.

Each of the two unbalances is driven by a separate electric motor. The options for the mutual arrangement of the unbalances are shown in Figure 1, where *O* is the axis of rotation, m is the weight of the unbalance, *ω* is the angular velocity of rotation, *α* is the angle that determines the relative position of the unbalances, *F* is the disturbing force, *F_max_* is the maximum value of the disturbing force.

In the first case (Figure 1a), the unbalances rotate in the antiphase, and the disturbing force is zero. In the second case (Figure 1b), the unbalances rotate in phase, and the disturbing force is maximal. In the third case (Figure 1c), the disturbing force is determined by the expression:(1)$$F=F_{\max}\cdot \cos(\frac{\alpha }{2}),\begin{array}{cc} & \alpha \in [0,\pi ] \end{array},$$

The variants in Figure 1a,b are essentially special cases of the variant in Figure 1c.

For nonregulated vibrating feeders, a pair of unbalance selectors is used, the rotors of which rotate in the opposite directions, forming an alternating stirring force acting along the axis of the perpendicular segment connecting the axes of rotation of the unbalances and passing through its middle. In this case, the mutual arrangement of the unbalances is provided by the phenomenon of self-synchronization.

The regulation of the mutual position of the two unbalances, the axes of which are parallel, as proposed in [15,16], leads to the creation of an additional torque, which has a negative effect on the vibrating unit.

The use of two pairs of unbalanced vibrators allows, by changing the mutual angular position of one pair relative to the other, to adjust the amplitude of the disturbing force [15,37].

Let us consider a system consisting of 4 unbalanced vibration drives (Figure 2). For the simplicity of exposition, we will assume that the masses m and eccentricities *ε* of all unbalances are the same, and their axes of rotation *O*1–*O*4 are located symmetrically to the origin of coordinates *O*.

The projections ($F_{x},\;F_{y}$) of the normalized disturbing force $F^{*}=F/(m\cdot \varepsilon \cdot \omega^{2})$ are determined by the following expressions, where $\varphi_{1},\varphi_{2},\varphi_{3},\varphi_{4}$ are the unbalance angles:(2)$$\{\begin{array}{c} F_{x}^{*}=\cos\varphi_{1}+\cos\varphi_{2}+\cos\varphi_{3}+\cos\varphi_{4};\\ F_{y}^{*}=\sin\varphi_{1}+\sin\varphi_{2}+\sin\varphi_{3}+\sin\varphi_{4}.\;\;\;\; \end{array}$$

Managing the movement of unbalances in accordance with the following dependencies:(3)$$\left\{ \begin{array}{l} \varphi_{1}=\omega t+\beta +\gamma ;\\ \varphi_{2}=\omega t-\beta +\gamma +\pi ;\\ \varphi_{3}=-\omega t-\beta \;;\\ \varphi_{4}=-\omega t+\beta +\pi , \end{array}\right.$$
where ω is the angular frequency of the rotation of unbalances and β and γ are the variable parameters that determine the relative location of the unbalances; we obtain:(4)$$\{\begin{array}{c} F_{x}^{*}=4\sin\beta \cos\frac{\gamma }{2}\sin\left(\omega t+\frac{\gamma }{2}+\pi \right);\\ F_{y}^{*}=4\sin\beta \sin\frac{\gamma }{2}\sin\left(\omega t+\frac{\gamma }{2}+\pi \right). \end{array},$$

As can be seen, the parameter β determines the amplitude of the disturbing force, and the parameter γ determines its direction. 

Thus, by controlling the angular positions of the four unbalanced electric drives in accordance with the dependencies (3), it is possible to separately control the frequency, amplitude, and direction of the vibration. The functional scheme of the four unbalanced vibration exciters is shown in Figure 3, where *f, A_a_,* and Θ are the specified frequency, amplitude, and direction of the vibration; *M*_1_ … *M*_4_ are the engine torques; *UV*_1_
*…UV*_4_ are the unbalanced vibrator; *AS* is the accelerometer (acceleration sensor); $\varphi_{1}\ldots \varphi_{4}$ are the unbalance angles; and $\bar{a}$ is the acceleration of the unbalanced vibration exciter. 

It should be noted that all unbalanced vibrators, UV1–UV4, are located on a common platform that moves with variable acceleration, as a result of which additional moments appear relative to the axes of the rotation of the unbalances; that is, not only the rotating unbalances affect the moving platform, but the platform also affects them. Thus, the system is interconnected and nonlinear.

Let us consider the mechanical processes in an adjustable unbalanced vibration exciter with a plane-parallel motion of a movable platform.

The position of the movable platform (Figure 4) in the nonmoving coordinate system is given by the coordinates of its center of mass, $x_{pl,}y_{pl}$ (point A), and the angle of rotation relative to the specified center of mass, $\varphi_{pl}$. The position of the unbalance rotation axis (point B) is specified in the moving coordinate system $x_{m}y_{m}$, the axes of which are rigidly connected to the platform, and the origin of the coordinates is at point A. $x_{p}$ and $y_{p}$ are the coordinates of a point in a moving coordinate system.

In general, when several rotating unbalances act on the platform, its movement is described by the following system of equations:(5)$$\left\{ \begin{array}{l} m_{pl}{\ddot{x}}_{pl}=\sum F_{x}-k_{x}{\dot{x}}_{pl}-c_{x}x_{pl}\\ m_{pl}{\ddot{y}}_{pl}=\sum F_{y}-k_{y}{\dot{y}}_{pl}-c_{y}y_{pl}\\ J_{pl}{\ddot{\varphi }}_{pl}=\sum M-k_{\varphi }{\dot{\varphi }}_{pl}-c_{\varphi }\varphi_{pl} \end{array}\right.,$$
where $m_{pl}$ is the platform weight; $x_{pl}$ and $y_{pl}$ are the platform’s center of gravity coordinates; $J_{pl}$ is the platform’s moment of inertia; $\varphi_{pl}$ is the platform’s angle of rotation relative to the center of gravity; $F_{x}$ and $F_{y}$ are the projection of the disturbing force caused by the rotational movement of a separate unbalance; *M* is the moment of disturbing the unbalance force relative to the platform’s gravity center; $k_{x}$, $k_{y}$, and $k_{\varphi }$ are the viscous friction coefficients; $c_{x}$, $c_{y}$, $c_{\varphi }$—stiffness coefficients. 

The rotational movement of the unbalance has the form:$$J_{ub}{\ddot{\varphi }}_{ub}=M_{e}+M_{v}-k_{ub}{\dot{\varphi }}_{ub},$$
where $J_{ub}$ is the unbalance moment of inertia; $\varphi_{ub}$ is the unbalance angle; $M_{e}$ is the engine torque; $M_{v}$ is the vibration moment; and $k_{ub}$ is the unbalanced viscous friction coefficient.

The position of the platform point in the fixed coordinate system can be found from such a system:(6)$$\left\{ \begin{array}{l} x=x_{pl}+x_{p}\cos\varphi_{pl}-y_{p}\sin\varphi_{pl}\\ y=y_{pl}+y_{p}\cos\varphi_{pl}-x_{p}\sin\varphi_{pl} \end{array}\right.,$$
where $x_{p}$ and $y_{p}$ are the coordinates of a point in a moving coordinate system. In this case, the projections of the velocity and acceleration of the point are described by the following systems of equations:(7)$$\left\{ \begin{array}{l} V_{x}={\dot{x}}_{pl}-{\dot{\varphi }}_{pl}(-x_{p}\sin\varphi_{pl}-y_{p}\cos\varphi_{pl})\\ V_{y}={\dot{y}}_{pl}-{\dot{\varphi }}_{pl}(-y_{p}\sin\varphi_{pl}+x_{p}\cos\varphi_{pl}) \end{array}\right.,$$
for speed, and
(8)$$\left\{ \begin{array}{l} a_{x}={\ddot{x}}_{pl}-{\ddot{\varphi }}_{pl}(-x_{p}\sin\varphi_{pl}-y_{p}\cos\varphi_{pl})\\ +{\dot{\varphi }}_{pl}{}^{2}(-x_{p}\cos\varphi_{pl}+y_{p}\sin\varphi_{pl})\\ a_{y}={\ddot{y}}_{pl}-{\ddot{\varphi }}_{pl}(-y_{p}\sin\varphi_{pl}+x_{p}\cos\varphi_{pl})\\ +{\dot{\varphi }}_{pl}{}^{2}(-y_{p}\cos\varphi_{pl}-x_{p}\sin\varphi_{pl}) \end{array}\right.,$$
for acceleration. 

The vibration moment resulting from the movement of the unbalance rotation axis [14] is found according to the formula:(9)$$M_{v}=m_{ub}\varepsilon (a_{x}\sin\varphi_{ub}-a_{y}\cos\varphi_{ub}),$$
where $m_{ub}$ is the mass of unbalance; ε is the eccentricity; and $\varphi_{ub}$ is the unbalance angle.

The impact of the unbalance on the moving platform is determined by the following system of equations:(10)$$\left\{ \begin{array}{l} F_{x}=m_{ub}\varepsilon {\dot{\varphi }}_{ub}{}^{2}\cos\varphi_{ub}\\ F_{y}=m_{ub}\varepsilon {\dot{\varphi }}_{ub}{}^{2}\sin\varphi_{ub}\\ M=F_{x}x_{p}-F_{x}y_{p} \end{array}\right.,$$
where $F_{x}$ and $F_{y}$ are the perturbing force projections; and M is the rotational moment relative to the platform’s center of gravity.

To implement a closed control system, the functional diagram of which is shown in Figure 3, information is needed on the relative position of the rotating unbalances (*φ*_1_–*φ*_4_). It should be noted that the existing sensors have a relatively high cost, low vibration resistance, and significant difficulties in mechanical coupling with the unbalance shaft due to the design features of the vibration units, which practically exclude the use of such sensors. 

Taking into account the features of unbalanced vibration drives, namely, that the angular velocity is not zero, changes slightly during one turnover, and is within known limits, a method for measuring the mismatch of the angular position of the rotating unbalances [27] was proposed, the distinguishing element of which is a coding disk with a missing tooth. This method allows to determine the angular position of the unbalance on the basis of a significant difference in the period of the pulse formed by the missing tooth in relation to the periods of the other pulses.

The delay of the measured value of the speed ($\tau_{\omega }$) and the angular mismatch of the unbalances relative to their real values ($\tau_{\Delta \varphi }$) is determined by the expressions:(11)$$\tau_{\omega }=\frac{2\pi }{\omega_{\mathrm{ub}}\cdot N_{t}},$$
where $N_{t}$ is the number of teeth on the encoder disc.
(12)$$\tau_{\Delta \varphi }=\frac{2\pi }{\omega_{\mathrm{ub}}},$$

Under the condition of the synchronous and uniform rotation of the unbalances, the use of sensors with a missing tooth allows to determine the angular mismatch of the unbalances using one signal from the sensor, unlike standard sensors with two output signals, which leads to a decrease in the cost of the system.

In the mathematical model of the control system of four unbalanced vibration exciters [17] in the speed control loop, the parameters of the frequency converter and the motor were not taken into account, and it was assumed that it was possible to set the required torque on the motor shaft in steps if there was a speed error.
(13)$$M_{e}(\delta_{\omega })=\left\{ \begin{array}{l} M_{r},\;\;\;\;\;\;\;\;if\;\;\;\;\delta_{\omega }>0\\ 0\;,\;\;\;\;\;\;if\;\;\;\;\delta_{\omega }=0\\ -M_{r},\;\;\;\;if\;\;\;\;\delta_{\omega }<0 \end{array}\right.,$$
where $\delta_{\omega }$ is the unbalance angular velocity error, and $M_{r}$ is the value of the switching moment of the relay speed controller.

## 3. Results

In this paper, the simulation model was supplemented with a description of a frequency-controlled asynchronous electric drive. The simulation model of an asynchronous motor (AM) based on the simplified Kloss formula in the MATLAB Simulink software environment is shown in Figure 5. To take into account electromagnetic transient processes, it was proposed to use an aperiodic link of the first order. For AM *P* = 1.1 kW, *f* = 50 Hz, *cos* = 0.78, *n* = 1390 rpm, and *U* = 400 V, the values of the parameters of the mathematical description are as follows: *M_cr_* = 2.4·7.55 H∙m, absolute slip corresponding to the critical moment *Δ**n_cr_* = 400 rpm, and synchronous speed *n*_0_ = 1500 rpm. In order to confirm the adequacy of the proposed mathematical description of the electric drive, mathematical modeling and a physical experiment were carried out for a motor controlled by a frequency converter, while the frequency reference of the converter changed every 0.5 s for two fixed values—30 and 32 Hz. 

A function ode45 implementing the Runge–Kutta method with a variable time step was used for modeling in MATLAB. The largest acceptable solver error, relative to the size of each state during each time step is 1 × 10^−3^. The combined Runge–Kutta (4–5) order method with automatic step selection is widely used to solve nonstiff systems of differential equations. This method satisfies the requirements for solving the system of differential equations considered in the study.

The block diagram of the simulation model of an asynchronous electric drive with a variable jump in setting the power supply frequency is shown in Figure 6, where $f_{g}$ is the given frequency; AM is the asynchronous motor; $M_{rez}$ is the resistance moment; and *C* is the speed encoder coefficient.

The block diagram of the stand for conducting the physical experiment is shown in Figure 7, where $f_{g}$ is the given frequency, FC is the frequency converter Sinamics G120 PM240, AM is the asynchronous motor, SS is the speed sensor, encoder Autonics E40H12-1000-3-T-24, and Osc. is the oscilloscope Rigol DC1042CD2+16.

During the physical experiment, pulses with a period of 1 s and a duty cycle of 50% were applied to the input of the frequency converter. The upper level corresponded to a rotation frequency of 16 rps, and the lower level corresponded to 15 rps. The three-phase variable-frequency power supply voltage from the output of the converter was fed to the asynchronous motor, the shaft of which, in turn, was connected to the encoder. The signal from the encoder was recorded using an oscilloscope in the form of an array. Further, the received data were processed in Excel in order to determine the time intervals during which 100 encoder pulses were received. Based on the obtained values, the instantaneous speed of the motor shaft was determined. The results of mathematical modeling and physical experiment are shown in Figure 8 and Figure 9.

The closeness of the results obtained provides grounds for using this mathematical model of AM in a closed system for controlling the angular position of the unbalances of an adjustable vibration exciter. The results of modeling the process of regulating the amplitude of a four-unbalanced vibration exciter using the mathematical model proposed in [17] supplemented by the model of a frequency-controlled asynchronous electric drive are shown in Figure 10.

In the process of modeling, in the interval 0–1 s, the launch was carried out at a given synchronous angular velocity, $\omega_{s}$ = 15.7 s^−1^, and parameter, β = 0°. After, the acceleration took place during which the given synchronous angular velocity, $\omega_{s}$, increased to 157 s^−1^. Then, at a constant $\omega_{s}$, parameter β within 9 s linearly increased from 0 to 180°. 

In this case, the engine power averaged over the period of rotation of the unbalances was determined in accordance with the expression:(14)$$P_{}=\frac{1}{\Delta t}\int_{t}^{t+\Delta t}M_{e}\omega_{ub}dt,$$
where Δt is the time of one unbalance revolution; $\omega_{ub}$ is the unbalance rotation speed.

## 4. Discussion

Thanks to the unique technological possibilities, vibration exciters have found wide application in a fairly large number of branches of light and important industry and agriculture. The most common are electromagnetic and imbalance vibration exciters. They have both advantages (the possibility of adjusting the vibration amplitude by changing the supplied electricity) and disadvantages (high material consumption).

To ensure the stability of the vibration installation to the influence of the technological load, the mechanical power of the vibration exciter is overestimated. Difficult starting conditions of the vibration installation also affect the increase in the power of the vibration exciter during its design. These factors have a negative impact on the energy efficiency of unbalanced vibratory installations. Regulating the amplitude of the disturbance force of unbalanced vibration exciters, regardless of the vibration frequency, will allow to reduce the set power of the unit due to the transmission of the resonant frequency with a minimum disturbance force and compensation of the impact of the loading process using a closed electric drive.

One of the ways to combine the advantages of both types of vibration exciters, namely, low material consumption and the ability to adjust the disturbing force during work, is the use of four mass-produced unbalanced motors placed on the same platform in combination with a frequency electric drive.

As can be seen from Figure 1 and Figure 2 and Formulas (1)–(4), provided that the movement of each individual imbalance is independently controlled, it is possible to separately control the frequency, amplitude, and direction of the disturbing force.

At the same time, placing imbalances on a common platform entails the mutual influence of imbalances on each other. This feature makes the system interconnected and nonlinear. The mechanical processes of the interaction of the imbalances rotating around axes with a common platform and the laws of platform motion are described by Formulas (5)–(10). In this paper, the simulation model was supplemented with a description of a frequency-controlled asynchronous electric drive. The simulation model of an asynchronous motor based on the simplified Kloss formula is shown in Figure 5. These dependencies, together with the model of a frequency-controlled asynchronous motor, the adequacy of which was verified by a physical experiment (Figure 7 and Figure 8), are embodied in a simulation model. 

A physical experiment was conducted to determine the transient processes of changing the angular speed of an asynchronous motor with a sudden change in the setting of the frequency converter. On the basis of this experiment, the previously created mathematical model for the description of the dynamic parameters of the electric drive was refined. The proposed structure of the control system, the operability of which is confirmed by mathematical modeling, makes it possible to implement an adjustable four-imbalanced vibration exciter using single serially available asynchronous vibrators. 

The simulation results show that the use of a four-way unbalanced frequency-controlled vibration exciter allows to pass through the resonance zone with an amplitude significantly smaller than the amplitude in the operating mode, which will have a favorable effect on the duration of the operation of the vibratory units due to the reduction of the dynamic loads on their nodes and parts in the process start and stop.

## 5. Conclusions

The use of a closed-loop frequency-controlled electric drive for four unbalanced vibration exciters makes it possible to regulate the oscillation amplitude directly during the operation of the vibrating unit. 

The proposed control method, which consists of a stepwise change in the setting for the frequency converter, allows to maintain a given speed of rotation of the unbalances and their relative position.

The proposed mathematical model, which takes into account the static and dynamic characteristics of an asynchronous frequency electric drive and the delay of the position feedback sensor, makes it possible to study the interaction of electric drives of individual unbalances due to the plane-parallel movement of the movable platform and can be used to develop a control system for an adjustable vibration exciter.

For AM *P* = 1.1 kW, *f* = 50 Hz, *cos* = 0.78, *n* = 1390 rpm, and *U* = 400 V, the values of the parameters of the mathematical description are as follows: *M_c_*_r_ = 2.4·7.55 H∙m, absolute slip corresponding to the critical moment *Δ**n_c_*_r_ = 400 rpm, and synchronous speed *n*_0_ = 1500 rpm.

In order to confirm the adequacy of the proposed mathematical description of the electric drive, mathematical modeling and a physical experiment were carried out for a motor controlled by a frequency converter, while the frequency reference of the converter changed every 0.5 s for two fixed values—30 and 32 Hz. 

Improving the energy efficiency of vibration machines is possible only with a comprehensive consideration of the energy conversion processes in the electromechanical complex “regulated electric drive–vibration exciter–vibration unit–technological load”.

The search for the stability conditions for the synchronous rotation of the mechanisms of vibration machines, provided by a closed controlled electric drive, taking into account the influence of the technological load, is a promising direction in the development of the theory of vibration machines and an automated electric drive.

## Figures and Tables

**Figure 1 sensors-23-02170-f001:**
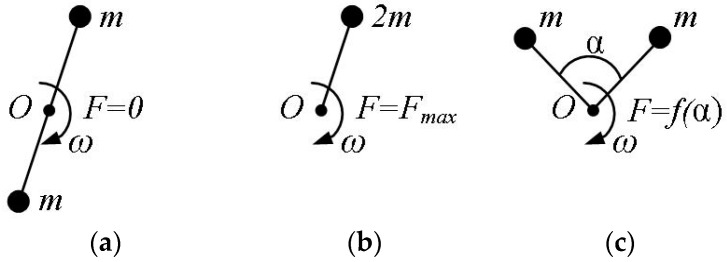
Regulation of the disturbing force. (**a**) the unbalances rotate in the antiphase, and the disturbing force is zero; (**b**) the unbalances rotate in phase, and the disturbing force is maximal; (**c**) the disturbing force.

**Figure 2 sensors-23-02170-f002:**
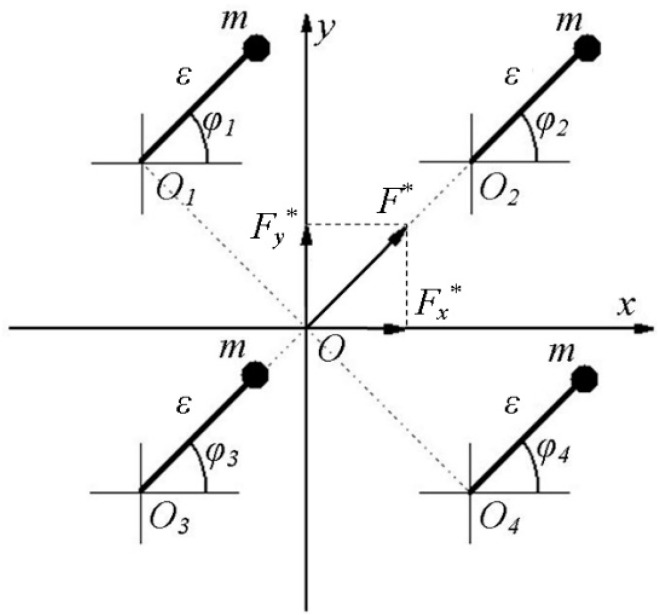
A four-unbalanced vibration exciter.

**Figure 3 sensors-23-02170-f003:**
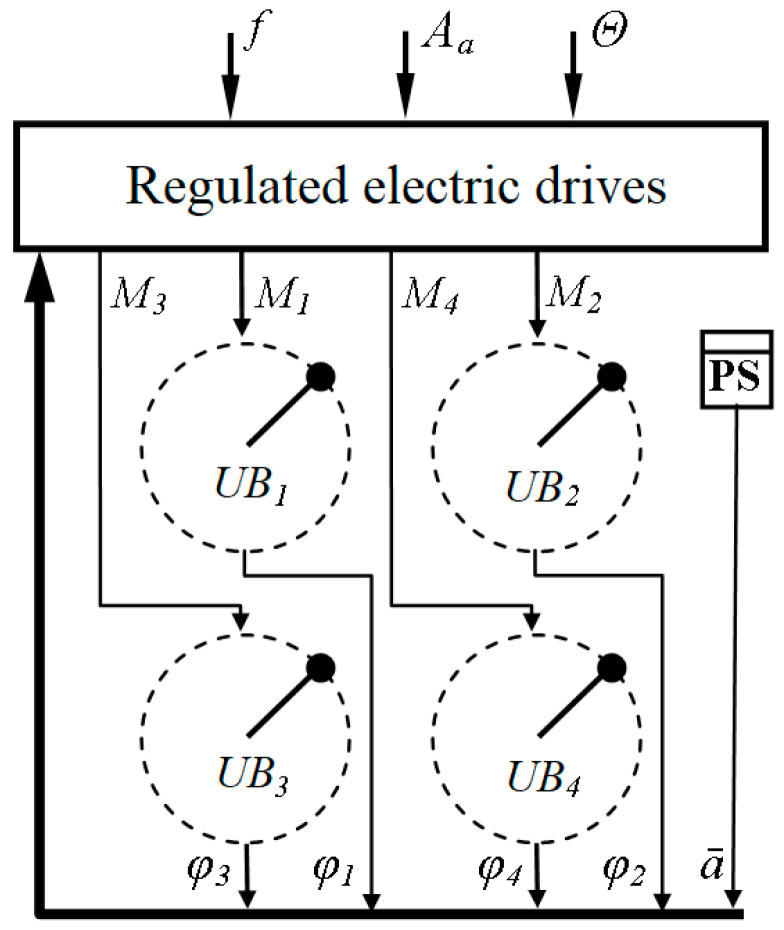
Functional scheme of the vibration exciter.

**Figure 4 sensors-23-02170-f004:**
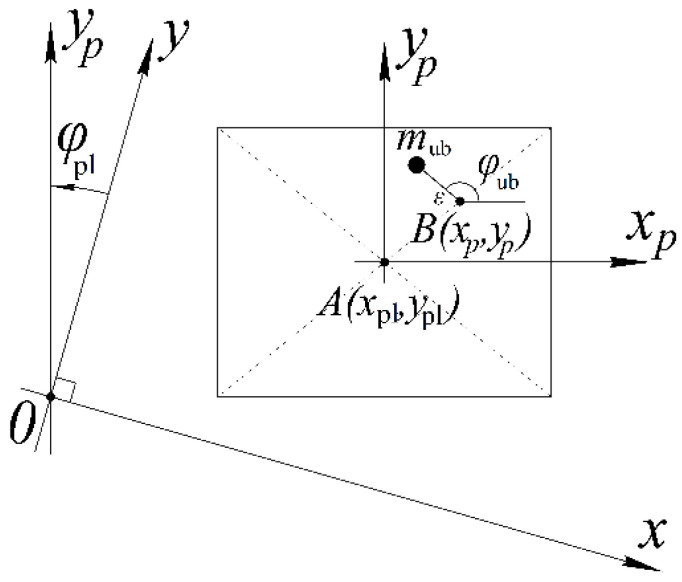
Unbalance on the moving platform.

**Figure 5 sensors-23-02170-f005:**
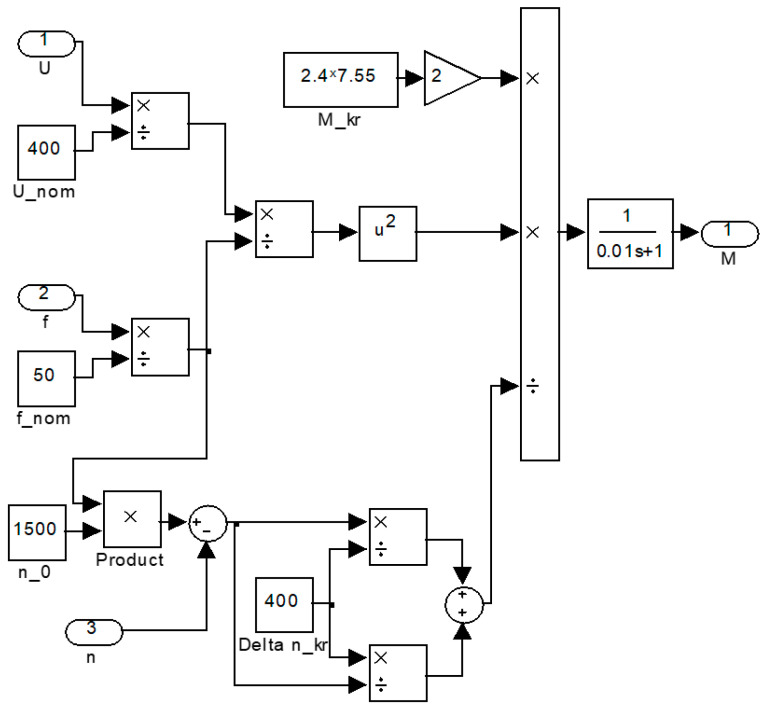
Model of a frequency-controlled AM.

**Figure 6 sensors-23-02170-f006:**
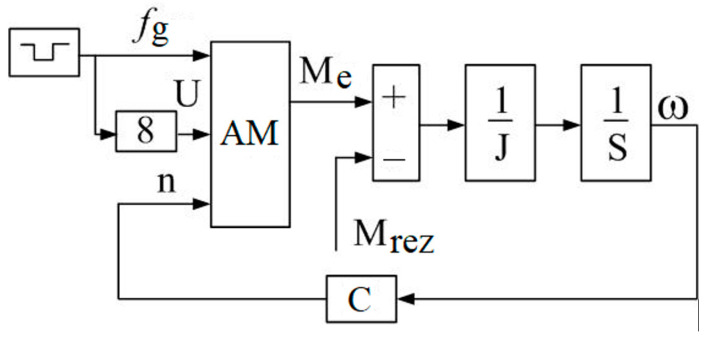
Structural diagram of the AM simulation model.

**Figure 7 sensors-23-02170-f007:**
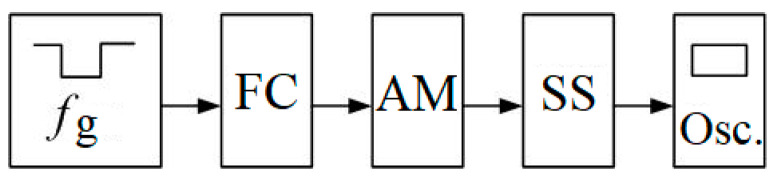
Structural scheme of the experimental stand.

**Figure 8 sensors-23-02170-f008:**
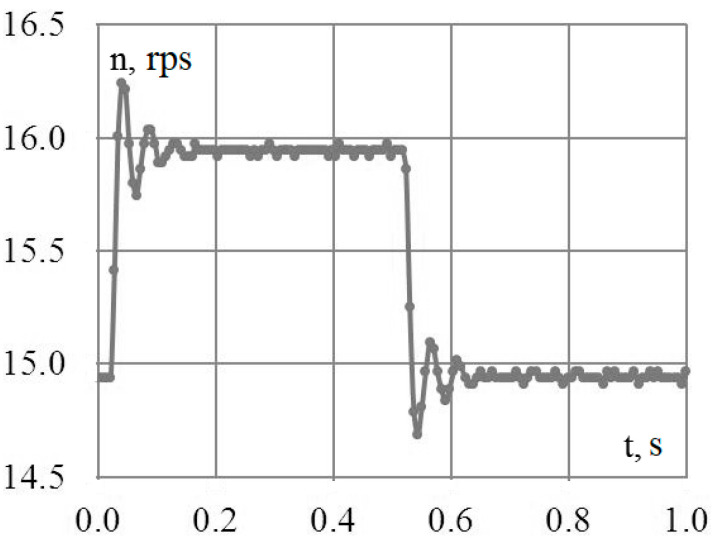
Results of the physical experiment.

**Figure 9 sensors-23-02170-f009:**
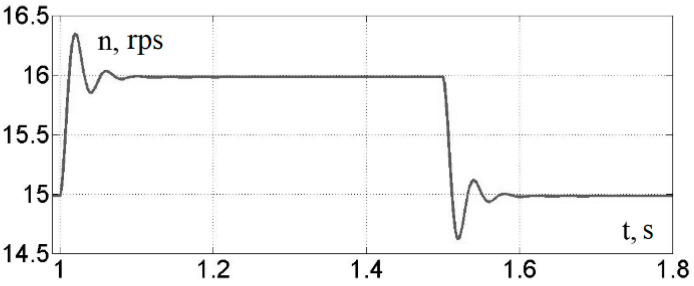
Results of the mathematical experiment.

**Figure 10 sensors-23-02170-f010:**
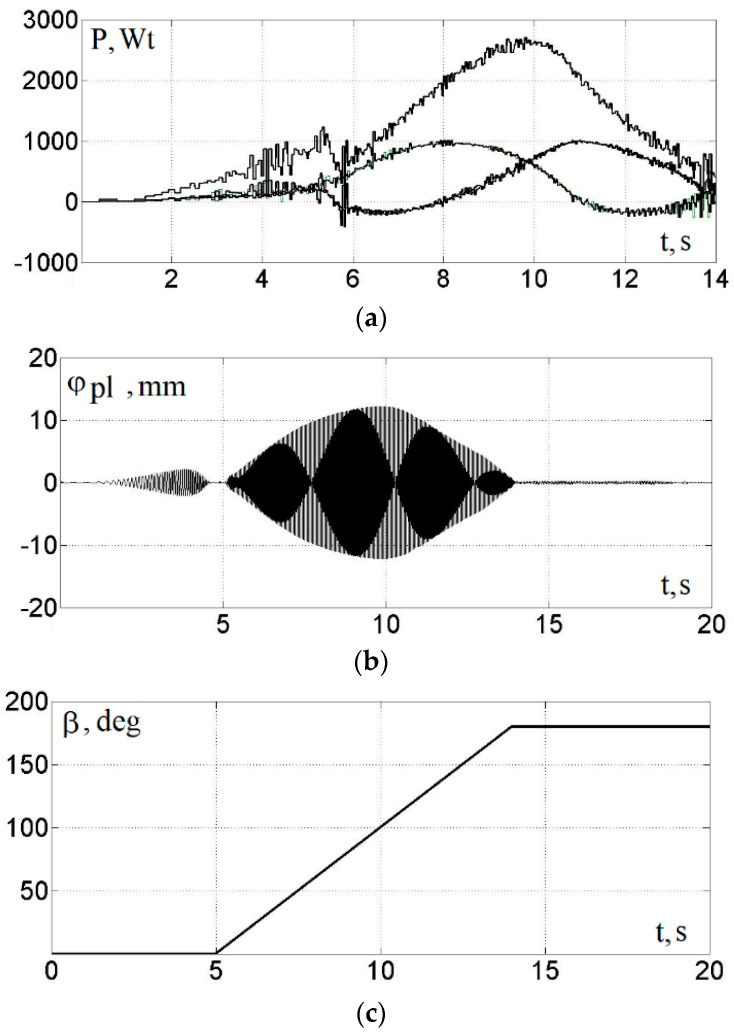
Graphs of the engine power when adjusting the amplitude of the platform oscillations: power of unbalance drive motors, Wt (**a**); platform position along the x-axis, mm (**b**); parameter value β, degrees (**c**); angular velocity of unbalances, s^−1^ (**d**).

## Data Availability

Not applicable.

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
