# Peer review of "Adjustable Vibration Exciter Based on Unbalanced Motors"

_sensors, 2023, doi:10.3390/s23042170_

Round 1
Reviewer 1 Report
The manuscript " Adjustable Vibration Exciter Based on Unbalanced Motors" has been reviewed, and I have the following comments:
1.The review section should be supported by more references' titles recently published (2020-2022).
2. The introduction should contain, in the end, a brief scheme of the paper sections. Please insert it before developing the arguments for each section.
3. I recommend adjusting the keywords in order to enhance the visibility and searchability of the manuscript.
4. The English need more editing for grammatical errors and style.
6. Please check the symbols adopted in the manuscript. The nomenclature list must be added or more clearly specify each of the variables used.
7. The figures in the paper are low-quality pictures. The authors should enhance the dpi of these images to a higher print quality level (Ex. Fig. 9).
8. Please try to argue better the best behavior of your technique with respect to others already present in the literature. You should add a Discussion paragraph after the Results paragraph where you put in evidence the goodness and originality of your proposal.
9. From the applied numerical method, what were the selection criteria? What is the convergence criterion for the simulations?
Author Response
Comments and Suggestions for Authors The manuscript " Adjustable Vibration Exciter Based on Unbalanced Motors" has been reviewed, and I have the following comments:
1.The review section should be supported by more references' titles recently published (2020-2022).
Added a links
- The introduction should contain, in the end, a brief scheme of the paper sections. Please insert it before developing the arguments for each section.
Added a description of the sections
- I recommend adjusting the keywords in order to enhance the visibility and searchability of the manuscript.
Added 2 keywords
- The English need more editing for grammatical errors and style.
Edited
- Please check the symbols adopted in the manuscript. The nomenclature list must be added or more clearly specify each of the variables used.
Edited (2), (9), (11)
- The figures in the paper are low-quality pictures. The authors should enhance the dpi of these images to a higher print quality level (Ex. Fig. 9).
We looked at the drawings, when the image is enlarged, they retain clarity, the text of the signatures and numbers meets the requirements, they are large enough, and easy to read.
- Please try to argue better the best behavior of your technique with respect to others already present in the literature. You should add a Discussion paragraph after the Results paragraph where you put in evidence the goodness and originality of your proposal.
The "Discussion" section is located after the "Results" section
- From the applied numerical method, what were the selection criteria? What is the convergence criterion for the simulations?
Line 323-328
A function ode45 implementing the Runge-Kutta method with a variable time step was used for modeling in Matlab.The largest acceptable solver error, relative to the size of each state during each time step is 1e-3. The combined Runge–Kutta (4–5) order method with automatic step selection is widely used to solve non-stiff systems of differential equations. This method satisfies the requirements for solving the system of differential equations considered in the study.
Reviewer 2 Report
The article proposes the adjustable four-unbalanced vibration exciter. The scientific novelty of the scientific research is in the proposed reliable mathematical model of the comprehensive mechanical system “electric drive – exciter – vibration unit – loads”. The practical significance of the obtained results is in the possibility of improving the energy efficiency of vibration machines.
However, despite the undoubtful advantages of the proposed methodology and the obtained results, the following minor revision is needed due to the following comments:
1. The application to only the Ukrainian industry is mentioned at the beginning of the abstract and introduction. However, the application of your study is more widespread due to the use of such types of adjustable vibration exciters worldwide. In this regard, I recommend extending the application by replacing it, at least in the European industry.
2. The title “2. Analysis of recent research and publications.” should be removed, and this chapter should be merged with the introduction. Therefore, other chapters should be renumbered.
3. Formula (2) contains the Chinese “Jiān / 牋” text and should be removed. Also, after this formula, the following clarification can be added: “This system considers projections of the specific centrifugal force F* = F/(m·ε·ω^2)”, or maybe something else in this manner.
4. The simulation model of an asynchronous motor should be extended by indicating the program/software of its numerical realization.
5. It is also unclear how the results of physical experiments have been obtained (more explanations are needed).
6. The manuscript contains an unacceptable level (11 items, 29%) of self-citations [5, 6, 7, 15, 23, 25, 26, 29, 30, 37, 38] that must be reduced to a maximum of 2-3 items.
Author Response
Comments and Suggestions for Authors.
The article proposes the adjustable four-unbalanced vibration exciter. The scientific novelty of the scientific research is in the proposed reliable mathematical model of the comprehensive mechanical system “electric drive – exciter – vibration unit – loads”. The practical significance of the obtained results is in the possibility of improving the energy efficiency of vibration machines. However, despite the undoubtful advantages of the proposed methodology and the obtained results, the following minor revision is needed due to the following comments:
- The application to only the Ukrainian industry is mentioned at the beginning of the abstract and introduction. However, the application of your study is more widespread due to the use of such types of adjustable vibration exciters worldwide. In this regard, I recommend extending the application by replacing it, at least in the European industry.
Corrected
- The title “2. Analysis of recent research and publications.” should be removed, and this chapter should be merged with the introduction. Therefore, other chapters should be renumbered.
Merged and renumbered
- Formula (2) contains the Chinese “Jiān / 牋” text and should be removed. Also, after this formula, the following clarification can be added: “This system considers projections of the specific centrifugal force F* = F/(m·ε·ω^2)”, or maybe something else in this manner.
Formula (2) is missing Chinese characters, perhaps it is not read correctly by the reviewer in the text editor.
A clarification was added after the formula: “The projections of the normalized disturbing force F*= F/(m·ε·ω^2) are determined by the expressions.”
- The simulation model of an asynchronous motor should be extended by indicating the program/software of its numerical realization.
Added (line 312-313)
- It is also unclear how the results of physical experiments have been obtained (more explanations are needed).
Added extended description (line 344-352)
- The manuscript contains an unacceptable level (11 items, 29%) of self-citations [5, 6, 7, 15, 23, 25, 26, 29, 30, 37, 38] that must be reduced to a maximum of 2-3 items.
Fixed, left 3 self-citations.
Round 2
Reviewer 1 Report
I consider that the proposed improvements were implemented and that the manuscript can be accepted in its current version.